# Systematic variation of the acceptor electrophilicity in donor-acceptor-donor emitters exhibiting efficient room temperature phosphorescence suited for digital luminescence

Uliana Tsiko[1], Jannis Fidelius [2], Sebastian Kaiser[1], Heidi Thomas [1], Yana Bui Thi[1], Jan J. Weigand [2], Juozas V. Grazulevicius[3], Karl Sebastian Schellhammer [1,4] & Sebastian Reineke [1,4] ✉

Purely organic materials showing efficient and persistent emission via room temperature phosphorescence (RTP) allow the design of minimalistic yet powerful technological solutions for sensing, bioimaging, information storage, and safety applications using the photonic design principle of digital luminescence. Although several promising materials exist, a deep understanding of the underlying structure-property relationship and, thus, development of rational design strategies are widely missing. Some of the best purely organic emitters follow the donor-acceptor-donor design motif. In this study, the influence of the acceptor unit on the photophysical properties is systematically analyzed by synthesizing and characterizing variations of the RTP emitter 4,4′-dithianthrene-1-yl-benzophenone (**BP-2TA**). The most promising candidates are also tested in programmable luminescent tags as a potential application field for information storage. While no significant influence by the electrophilicity index of the acceptor moiety on the RTP emission is observed, the results support the design of molecules with pronounced hybridization as obtained for the newly synthesized emitter demonstrating superior RTP efficiency combined with improved stability.

Academic and industrial sectors have shown considerable interest in materials that exhibit persistent luminescence, characterized by extended emission lifetimes of microseconds and beyond. These materials have gained attention due to their wide-ranging advantages in applications across diverse fields, including molecular sensing, bioimaging, information storage, and anti-counterfeiting applications, as they combine performance with relatively easy read-out[1–11]. In addition to their optical properties, purely organic luminescent materials offer advantages such as solution processability, mechanical flexibility, compatibility with various substrates, and low ecological footprint. These characteristics make them suitable for large-scale manufacturing processes of flexible photonic devices.

Over the last years, we have developed and refined the technology of programmable luminescent tags (PLTs) for minimalistic and powerful,

reusable, industrially compostable information storage, labeling, and sensing[12–15]. PLTs are easily processed from solution and can be printed onto any substrate and in large quantities. Their architecture is schematically depicted in Fig. 1a. They consist of a polymer layer doped with a room temperature phosphorescence (RTP) emitter at a concentration of a few weight percent (wt%). This film is covered by a layer of Exceval (TM Kuraray Europe GmbH; modified ethylene-vinyl alcohol copolymers) as a switchable oxygen-blocking layer. According to this design, they allow for digital luminescence, i.e., the local control of a dark or bright state based on phosphorescence using oxygen-based quenching. Local activation of phosphorescence occurs by using a shadow mask and exciting the emitter molecules with UV light, which triggers a photoconsumption of the oxygen molecules. (cf. Fig. 1b, c). A reflux of oxygen is prohibited by the oxygen

[1]Dresden Integrated Center for Applied Physics and Photonic Materials (IAPP), Technische Universität Dresden, Dresden, Germany. [2]Chair of Inorganic Molecular Chemistry, Faculty of Chemistry and Food Chemistry, Technische Universität Dresden, Dresden, Germany. [3]Department of Polymer Chemistry and Technology, Kaunas University of Technology, Kaunas, Lithuania. [4]These authors jointly supervised this work: Karl Sebastian Schellhammer, Sebastian Reineke.
✉e-mail: Sebastian.reineke@tu-dresden.de

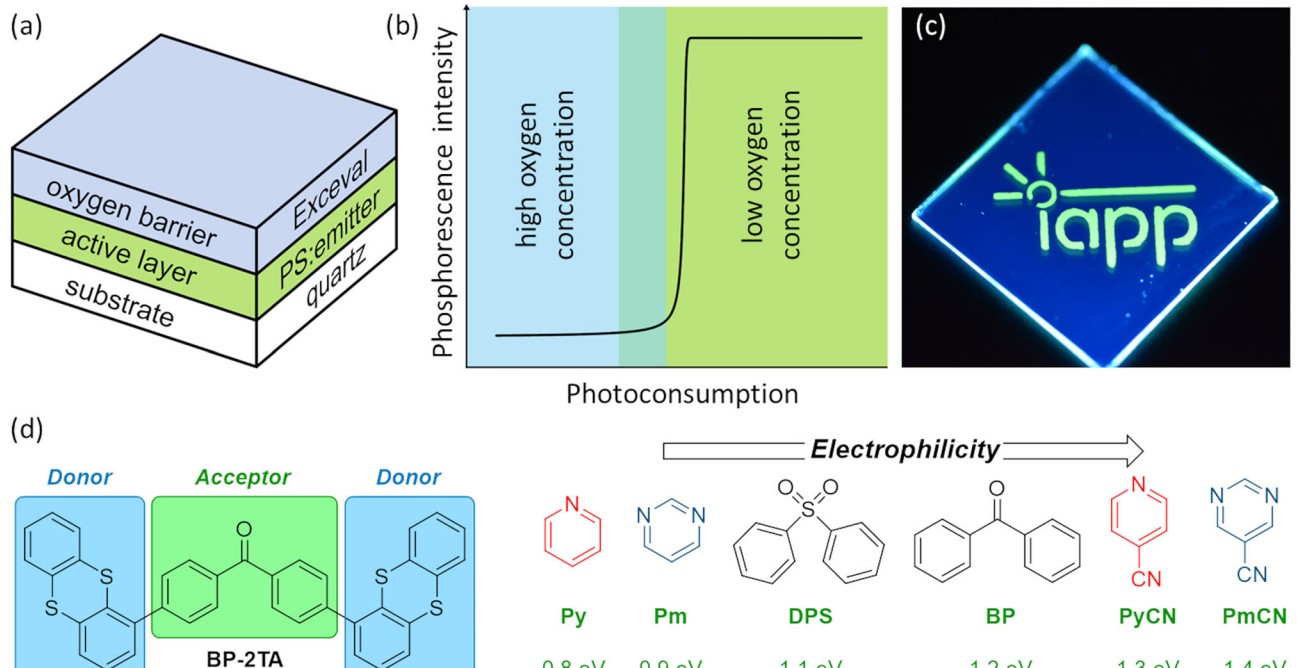

**Fig. 1 | Programmable luminescent tags (PLTs) and tested emitter designs.**
**a** Schematic architecture of a PLT. **b** Oxygen-controlled activation of the phosphorescent emission. Due to photoconsumption, molecular oxygen is consumed leading to bright persistent emission after a system-specific activation dose of UV light. **c** Locally activated PLT with the best emitter **Py-2TA** (5 wt% in PS). Activated areas with sufficiently low oxygen concentration show bright green phosphorescence while the remaining non-activated areas exhibit weak blue fluorescence. **d** Molecular design of the phosphorescent organic emitters studied. While thianthrene is always used as donor group, the acceptor unit is systematically varied covering a broad range of values of the electrophilicity index (CAM-B3LYP/6-311 G**) with pyrimidine (Pm), diphenylsulfone (DPS), and benzophenone (BP) having been realized elsewhere[19].

barrier. However, a deactivation of the RTP emission and, thus, an erasure of the information stored is possible by heating the PLT, resulting in an increase of the permeability of the barrier layer. Efficient PLTs allow for multiple writing and erasing cycles. However, the limited availability of cost-effective purely organic molecules with strong RTP emission that cover the entire wavelength spectrum prevents the exploiting of the full potential of this promising technology and the photonic design principle of digital luminescence.

Among other properties, the emitter should not only exhibit high phosphorescence quantum yield $\varphi_P$ but also low fluorescence quantum yield $\varphi_F$ to provide an excellent contrast for read-out, even while the sample is photo-excited and in bright-light settings. Especially when fluorescence and phosphorescence are not well separated in energy, the quality of an RTP emitter for PLTs can be characterized by the Weber contrast $K_W$[16]. It is defined as $K_W = \frac{L_{max}}{L_{min}} - 1$ with $L_{max}$ and $L_{min}$ the maximum and minimum luminance, respectively. In our case, the luminance $L$ is replaced by the number of emitted photons per unit time in nitrogen $L_{N_2}$ or air $L_{air}$ giving $K_W = \frac{L_{N_2}}{L_{air}} - 1$. In addition to its photophysical properties, the emitter should provide reasonable chemical and thermal stability as well as photostability, showing only weak decrease in intensity over writing and erasing cycles of PLTs. For example, the phosphorescence intensity decreases down to 40% after 40 cycles for the RTP emitter N,N′-di(1-naphthyl)-N,N′-diphenyl-(1,1′-biphenyl)-4,4′-diamine (NPB)[12], which may be insufficient for information storage applications.

So far, 4,4′-dithianthrene-1-yl-benzophenone (**BP-2TA**, cf. Fig. 1d) has demonstrated the best performance as RTP emitter in PLTs[13]. It follows the donor-acceptor-donor (D-A-D) design motif, which is well-known from thermally-activated delayed fluorescence (TADF) emitters[17,18], with thianthrene (TA) as donor and benzophenone (BP) as acceptor[19]. Compared to structurally similar D-A emitter molecules, the D-A-D pattern appears to provide improved thermal stability and higher glass transition temperature, which is especially relevant for application in vacuum-processed devices[19]. **BP-2TA** exhibits a pronounced green phosphorescence

with $\varphi_P$ up to 21% and a lifetime $\tau_P$ of 30 ms combined with almost negligible fluorescence, which is also well separated in energy (cf. Fig. S30)[13].

Further representatives following the D-A-D structural motif consisting of thianthrene as a donor are 4,4′-dithianthrene-1-yl-diphenylsulfone (**DPS-2TA**)[19] and 4,6-di(thianthren-1-yl)pyrimidine (**mTEPm**)[20] with the latter being simultaneously and, thus, independently discovered in our work discussed here and named as **Pm-2TA** in the following. In all three cases, the C-1 position of the thianthrene units is used to connect them to the acceptor moieties to potentially decrease the conjugation between them[19]. Despite their structural similarities, both molecules show strong variation in their emission properties, relevant for application in photonic devices such as PLTs. **DPS-2TA** exhibits pronounced fluorescence, with $\varphi_F$ being six times larger than that observed for **BP-2TA**, accompanied by an estimated reduced $\varphi_P$, rendering it unattractive for application in PLTs[19]. In contrast, **Pm-2TA** also shows more effective fluorescence than **BP-2TA**. PLQY values in nitrogen atmosphere as high as 33.5% have been reported for neat films, making it generally appealing[20]. A complex interplay between its chemical structure, including heavy atom effects and n-π* transition characteristics, pronounced vibrational spin-orbit coupling, and intermolecular interactions has been identified as the reason for its strong RTP. Interestingly, fluorescence and phosphorescence occur in a similar wavelength range for **Pm-2TA**, which is differing from the other two structurally similar emitters. These characteristics have led to its application in OLEDs with a maximum EQE of almost 8% using **Pm-2TA** neat films as emissive layers[20].

Already, these three materials demonstrate that a rational design of improved purely organic RTP emitters remains challenging. We extend this set of D-A-D RTP emitters using thianthrene as the donor moiety by systematically substituting the acceptor unit by functional groups of different electrophilicity index, as depicted in Fig. 1d. Pyridine is an even weaker acceptor than pyrimidine, while 4-pyridinecarbonitrile and pyrimidine-5-carbonitrile exhibit stronger electron-accepting characters. By combining material simulations and experimental characterization, we gain systematic

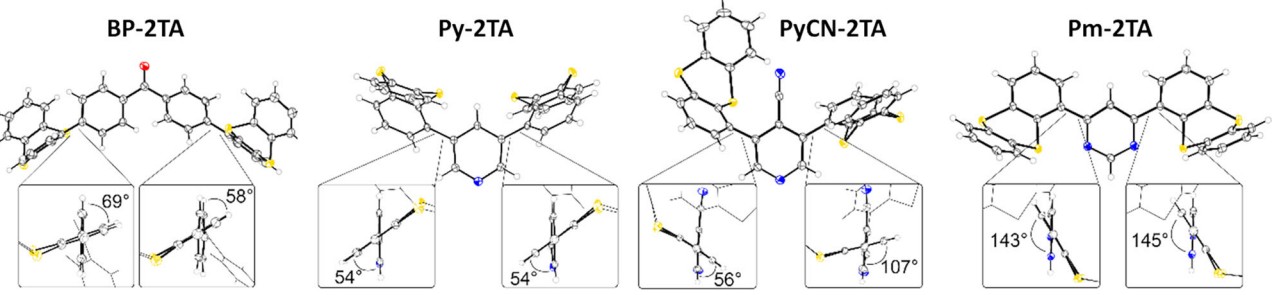

**Scheme 1 |** Synthetic routes for **Py-2TA**, **PyCN-2TA**, **Pm-2TA**, and **PmCN-2TA** and corresponding yield after purification.

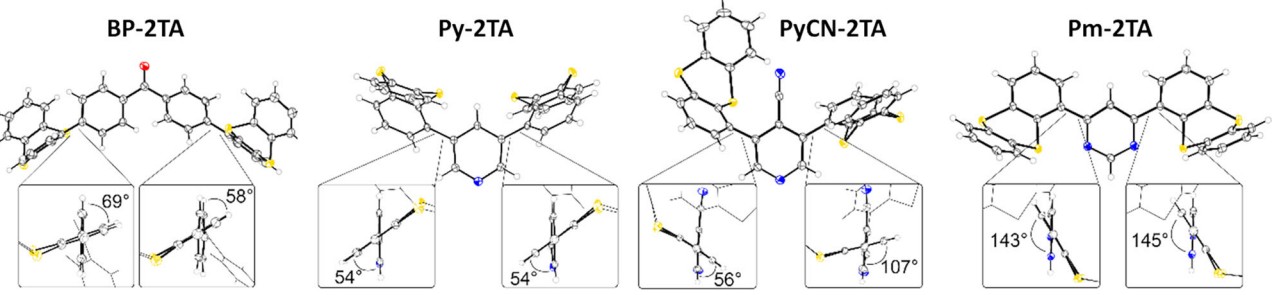

**Fig. 2 | Molecular structures of BP-2TA, Py-2TA, PyCN-2TA, and Pm-2TA.** Crystal structure information is provided in the SI. In terms of **Pm-2TA**, $CH_2Cl_2$ solvate molecules are omitted for clarity. Thermal ellipsoids are displayed at 50% probability.

insights in the interplay between chemical design, electronic structure, and photophysical properties. The electron-accepting character appears to not significantly modulate the emission properties. However, we find that pronounced hybridization might be favorable, as observed for the newly synthesized emitter with pyridine as the acceptor group. PLTs produced with this emitter show exceptionally bright green RTP emission with excellent contrast under illumination, due to a significantly weaker fluorescence (cf. Fig. 1c), which is combined with a convenient stability over the first ten writing and erasing cycles.

## Results and discussion
### Synthesis and crystal structures
The synthesis of the target compounds was conducted *via* Suzuki-Miyaura cross-coupling reactions. The synthetic routes for compounds: *3,5-di(thianthren-1-yl)pyridine* (**Py-2TA**), *3,5-di(thianthren-1-yl)-4-cyanopyridine* (**PyCN-2TA**), *4,6-di(thianthren-1-yl)pyrimidine* (**Pm-2TA**), and *4,6-di(thianthren-1-yl)pyrimidine-5-carbonitrile* (**PmCN-2TA**) are outlined in Scheme 1. The yields of reactions for compounds **Py-2TA, PyCN-2TA**, and **Pm-2TA** exceeded 60%, allowing for large-scale synthesis. The latter compound has been synthesized independently at the same time by the group of Shi-Jian Su following a similar yet slightly different synthetic route, resulting in an almost identical yield[20]. For **PmCN-2TA**, the yield is less than 10%, which is might be caused its low thermal stability.

The chemical compositions of all four compounds were confirmed by [1]H and [13]C NMR analysis and high-resolution mass spectrometry (see Section S1 in the Supplementary Information (SI) for more details). For **Py-2TA** and **PyCN-2TA**, single crystals were obtained by slow diffusion of *n*-hexane into an ethylacetate solution of the compounds, allowing for X-ray diffraction analyses. For comparison, the molecular structures of **BP-2TA** and **Pm-2TA** were also obtained and evaluated. In case of the latter

molecule, we obtained a different molecular structure than reported in ref. 20 due to a co-crystallization with DCM, also revealing a different molecular conformation. In general, all molecules show a similar preferred conformation with one thianthrene moiety pointing forward and one backward, as well as pronounced dihedral angles with respect to the acceptor group as shown in Fig. 2. The molecular structures of **BP-2TA** and **PyCN-2TA** reveal a significant asymmetry in the dihedral angles indicating a relatively small energy barrier for rotational motion of the thianthrene group around the connecting bond. This agrees with simulations reported for **Pm-2TA**[20]. It has been shown that a variety of conformations is accessible even under ambient conditions, strongly influencing the spin-orbit coupling and, thus, the statistics for electronic transitions. This includes conformers obtained from rotational motion as well as those resulting from a flipped attachment of the TA moiety, as observed in the two molecular structures of **Pm-2TA**.

### Electrophilicity index and electronic structure
To gain deeper insights into the electronic structure of the different emitters, density functional theory calculations were performed at the CAM-B3LYP/6-311 G**[21–23] level of theory. The electrophilicity index of the different acceptor moieties is presented in Fig. 1 to evaluate their electron-accepting character. As can be seen, the selected groups cover a relatively wide range. The acceptor moiety BP is in the middle, with **BP-2TA** as the reference molecule showing strong RTP and only weak fluorescence.

Although DPS has a similar electrophilicity index as BP, strongly different emission properties have been observed for **DPS-2TA**, with a much stronger fluorescence[19]. This already renders the electrophilicity index of the acceptor group as an essential tuning parameter for RTP emitters in question. Still, with the newly synthesized molecules covering a broad range, we aim to systematically address this objective.

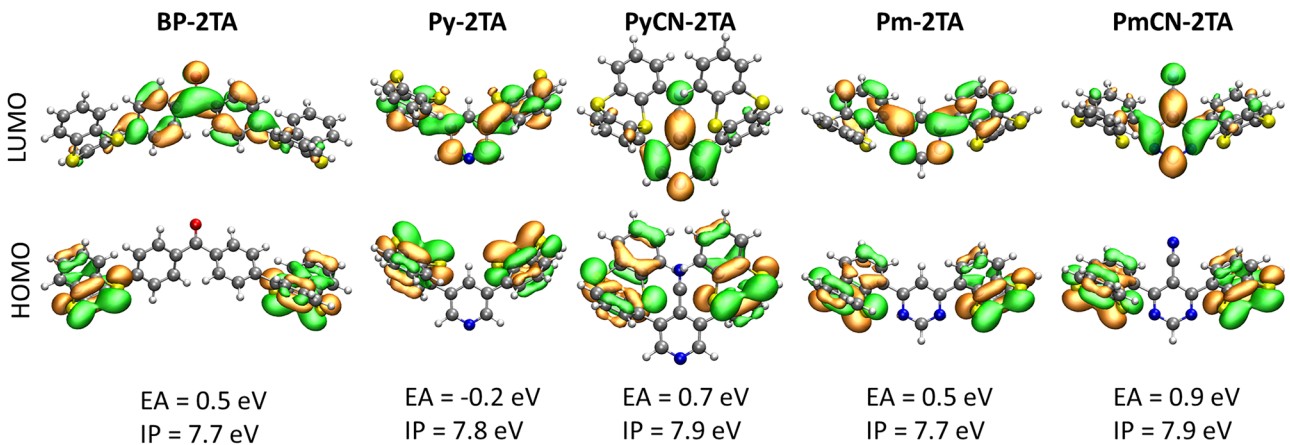

| BP-2TA | Py-2TA | PyCN-2TA | Pm-2TA | PmCN-2TA |

EA = 0.5 eV
IP = 7.7 eV

EA = -0.2 eV
IP = 7.8 eV

EA = 0.7 eV
IP = 7.9 eV

EA = 0.5 eV
IP = 7.7 eV

EA = 0.9 eV
IP = 7.9 eV

**Fig. 3 | Frontier molecular orbitals and energy levels of BP-2TA, Py-2TA, PyCN-2TA, Pm-2TA, and PmCN-2TA.** The dihedral angles between donor and acceptor moiety for the relaxed molecules in gas phase are 62°, 60°, 68°, 134°, and 125°, respectively. The relaxed geometries are given in Section S5 in the SI. The energy levels refer to gas phase values.

**Fig. 4 | Absorption spectra of BP-2TA, Py-2TA, PyCN-2TA, Pm-2TA, and PmCN-2TA.**
**a** Experimental spectra in PS. **b** Spectra simulated via TD-DFT. For comparison, the absorption of PS (**a**) and TA (**b**) are shown as well. The simulated spectra are obtained in gas phase and empirically red-shifted by 500 meV to approximately reproduce the experimental spectra. They are normalized to the maximum absorption intensity of **BP-2TA**, which shows the highest oscillator strength. For all compounds, absorption peaks with significant intensity correspond to transitions to higher excited singlet states (cf. Tabs S1-S5).

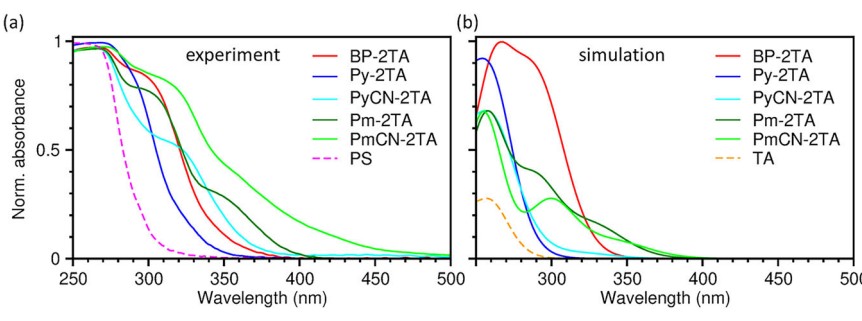

Where available, the molecular conformations obtained from single crystals were used as initial geometries for geometry relaxations in gas phase. For all molecules, they led to the relaxed molecular geometry with the global energy minimum as shown in Fig. 3. The conformation analyzed here for **Pm-2TA** is also energetically more favorable by 9 meV compared to the one studied in ref. 20, providing partially complementary results and, thus, offering a deeper understanding for this compound. The dihedral angles between donor and acceptor moiety are similar to the experimentally observed ones, but systematically increased, yielding a slightly more pronounced localization of the molecular orbitals. It should be noted that several rotamers exist for each molecule with only slightly higher energy, which are accessible under ambient conditions. Nevertheless, we expect the simulations to allow for qualitative comparison between the different materials regarding fundamental characteristics.

The frontier molecular orbitals of the full emitter molecules are depicted in Fig. 3. For **PyCN-2TA** and **PmCN-2TA**, a well-known charge transfer character is observed, with the highest occupied molecular orbital (HOMO) and the energetically very close HOMO-1 localized on the donor units and the lowest occupied molecular orbital (LUMO) localized on the acceptor unit. This matches the behavior of **BP-2TA**. In contrast, **Py-2TA** and **Pm-2TA** show a pronounced delocalization of the LUMO over the entire molecule, accompanied by smaller dihedral angles between donor and acceptor unit. In Fig. S21 in the SI, the frontier molecular orbitals of the individual moieties and their ionization potential (IP) and electron affinity (EA) in vacuum are presented. Surprisingly, the LUMO of **Py-2TA** corresponds to a hybridization of the LUMO of thianthrene and the LUMO + 1 of pyridine, although the LUMO of pyridine is energetically closer to the LUMO of thianthrene. It appears that this spectral shape favors a stabilization by hybridization, as it is similar to the LUMO of pyrimidine, which

forms the LUMO of **Pm-2TA** together with the LUMO of TA. We expect that the conjugation of the LUMO of both **Py-2TA** and **Pm-2TA** results from the energetic alignment of the EA of their donor and acceptor moieties. Due to their stronger electron-donating character, the EA of BP, PyCN, and PmCN is significantly increased, causing a stronger localization of the LUMO of the respective compounds.

The IP and EA of the full emitters in vacuum are presented in Fig. 3. In addition, an estimation in solution with DCM is given in Tab. 1 for comparison with cyclovoltammetry (CV) measurements. For **BP-2TA, PyCN-2TA**, and **PmCN-2TA** with a strong localization of HOMO and LUMO, their energy levels in vacuum are almost identical to those of their respective parts. This also applies to the IP of **Py-2TA** and **Pm-2TA**. The hybridization of the LUMO of **Pm-2TA** results in a stabilization of its EA rendering it almost similar to the EA of the other three compounds, despite reduced EA values of its individual units (cf. Fig. S21). For **Py-2TA**, the EA is strongly reduced due to the incorporation of the LUMO + 1 of pyridine, giving a fundamental gap that is increased by approximately 0.7 eV compared to the other compounds. Considering the simulated energy levels in DCM, these trends are widely conserved. Only **PyCN-2TA** reveals a further decrease in its fundamental gap due to a relatively large polarization shift. This might explain, why the experimental absorption spectrum is further redshifted compared to the simulated one (cf. Fig. 4).

## Thermal characteristics

In order to gain some understanding of the stability of the synthesized compounds, they were examined with simultaneous thermal analysis (STA). The results are presented in Section S3 in the SI. While the pyridine-based derivatives **Py-2TA** and **PyCN-2TA** show pronounced thermal stability with decomposition temperatures ($T_d$, obtained at 5% weight loss)

above 250 °C (see Table 1) and relatively high melting points above 150 °C, pyrimidine-based compounds **Pm-2TA** and **PmCN-2TA** behave differently, as they undergo an exothermic transformation around 125 °C, as observed in the respective DSC curves (see Figs. S19 and S20). This transformation is also responsible for a roughly 5% weight loss. Still, OLEDs with **Pm-2TA** have been successfully fabricated by physical vapor deposition[20], which indicates that the properties of the other materials are also sufficient for vacuum deposition.

### Photophysical characteristics

After their fundamental characterization, we discuss the photophysical properties of **Py-2TA**, **PyCN-2TA**, **Pm-2TA**, and **PmCN-2TA** and compare them with the characteristics of **BP-2TA** as a reference. Their absorption spectra in polystyrene (PS) at 5 wt% are depicted in Fig. 4a, and the spectra in solution with DCM are given in Fig. S27 in the SI. For **BP-2TA** and **Pm-2TA**, the same characteristic absorption features as published before were rediscovered[19,20]. In agreement with the largest fundamental gap, the absorption spectrum of **Py-2TA** is most blue-shifted in DCM and

PS. The fundamental gap of the remaining emitter molecules is similar leading to absorption in a similar wavelength range in solution. In contrast, in PS, **PmCN-2TA** exhibits relatively red-shifted absorption, which might be due to a variation in preferred molecular conformation. In general, all compounds show an extended absorption tail before their first absorption maximum, which might indicate relatively strong energetic disorder caused by a large variety of molecular conformations in the PS film.

For most molecules, the experimental spectra are consistent with the simulated ones presented in Fig. 4b. Only for **PyCN-2TA**, the shoulder around 320 nm could not be reproduced with the simulations. Consequently, they predict the $S_0 \rightarrow S_1$ transition to occur in this energy range, albeit with a relatively low oscillator strength. This might be the case only for the relaxed geometry. With the experimental spectrum being an ensemble average over a large variety of conformations, for some geometries, this transition might be connected to a notable absorption intensity. For **Pm-2TA**, the $S_0 \rightarrow S_1$ transition shows both a notable oscillator strength in the simulations and a distinct shoulder in the experimental absorption spectrum around 350 nm. For all other molecules, the $S_0 \rightarrow S_1$ transition can be considered optically inactive for absorption, which is also the case for the TA group itself. Accordingly, their absorption maxima correspond to transitions to higher excited singlet states. A detailed presentation of the individual transitions, including oscillator strength and natural transition orbitals (NTOs), is provided in Section S4 in the SI. Although the frontier molecular orbitals of **BP-2TA**, **PyCN-2TA**, and **PmCN-2TA** exhibit charge transfer character, NTOs of their most relevant excitations show pronounced hybridization between acceptor and donor moieties. According to the absorption spectra, illumination at 275 nm and 300 nm, as performed in the following, excites the molecules to higher singlet states.

The corresponding emission spectra of **Py-2TA**, **PyCN-2TA**, **Pm-2TA**, and **PmCN-2TA** diluted in PS under ambient and nitrogen atmosphere at room temperature are presented in Fig. 5 for $\lambda_{exc} = 275$ nm and in the SI in Fig. S28 for $\lambda_{exc} = 300$ nm[24]. The photophysical properties of **BP-2TA** are also summarized in Fig. S30 for comparison. Characteristic emission properties are given in Table 2. All compounds show pronounced RTP emission in the green with maximum wavelengths ranging from 508 to 530 nm, which is also visible in the delayed spectra (cf. Figs. 6a and S29a).

### Table 1 | Characteristics of BP-2TA, Py-2TA, PyCN-2TA, Pm-2TA, and PmCN-2TA

| Compound | $IP_{DCM}$[a], eV | $EA_{DCM}$[a], eV | $T_d$[b], °C | $T_m$[c], °C |
|---|---|---|---|---|
| **BP-2TA** | 6.6 (5.6[19]) | 2.3 (2.8[19]) | 442[19] | 209[19] |
| **Py-2TA** | 6.5 | 1.6 | 289 | 153 |
| **PyCN-2TA** | 6.2 | 2.4 | 322 | 214 |
| **Pm-2TA** | 6.6 | 2.2 | 170 | - |
| **PmCN-2TA** | 6.6 | 2.5 | 135 | - |

[a]Simulated energies of the frontier molecular orbitals in DCM as described in refs. 29,30.
Experimental values from cyclovoltammetry as reported in ref. 19 are given in parentheses revealing a quite strong deviation, potentially due to a variety of conformations available in solution. Still, it can be expected from former studies that the qualitative trends of the simulated results are well reflected allowing a comparison between the emitters.
[b]Temperature of 5% weight loss obtained from TGA.
[c]Melting temperature obtained from DSC.

**Fig. 5 | Emission spectra of Py-2TA, PyCN-2TA, Pm-2TA, and PmCN-2TA in PS.** Measurements were performed at room temperature under aerated (dashed lines, only fluorescence) and nitrogen atmosphere (solid lines, fluorescence and phosphorescence) with $\lambda_{exc} = 275$ nm.

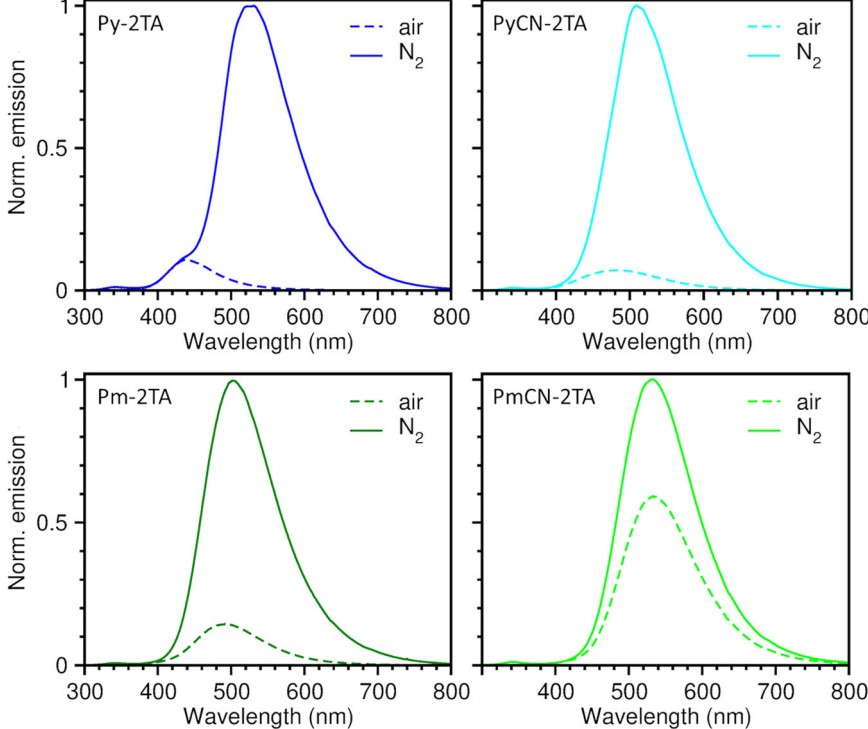

For **Pm-2TA** and **PmCN-2TA** the phosphorescence lifetime $\tau_P$ is barely longer than the delay time of the measurement setup leading to weak spectral signals. At 77 K, pronounced delayed spectra could be obtained for all emitter materials as shown in Fig. S31 due to further reduction of non-radiative processes. **Py-2TA** and **PyCN-2TA** reveal superior PLQYs of 25% and 24% under nitrogen atmosphere, respectively, exceeding the values obtained for the reference compound **BP-2TA**. While for **Pm-2TA** neat films, PLQY values as high as 33.5% have been reported[20], we observe values of up to 16% in PS, which is still as high as the PLQY of **BP-2TA**.

Regarding fluorescence, all emitters show diverse features. **BP-2TA** remains superior with very weak, blue-shifted fluorescence, giving the highest Weber contrast. However, **Py-2TA** and **PyCN-2TA** remain attractive due to a weak fluorescence, especially compared to their intense phosphorescence. **PmCN-2TA** exhibits the strongest fluorescence feature, making it unattractive for application in PLTs. **Pm-2TA** also appears less appealing than **Py-2TA** and **PyCN-2TA** because its fluorescence occurs at a wavelength close to its phosphorescence. Therefore, in the following, application in PLTs is only tested for **Py-2TA** and **PyCN-2TA** and compared with **BP-2TA**. In terms of the electrophilicity of the acceptor moiety, no systematic trend is observed. Although **Py** and **PyCN** differ strongly in their electron-accepting character and their electronic structure, their emission properties appear similar.

For the most promising emitter material **Py-2TA**, further photophysical analysis is performed to obtain an improved understanding. No significant alterations in the emission properties are obtained at lower (1 wt %) and elevated (10 wt%) emitter concentrations as depicted in Fig. S33. Fluorescence of **Py-2TA** remains almost unaffected in solutions of different

polarity, which is in stark contrast to **BP-2TA** as can be seen in Fig. S32. In agreement with the materials simulations, this indicates a less pronounced charge transfer character of the $S_1 \rightarrow S_0$ transition. It needs to be noted that these measurements revealed improved solubility of **Py-2TA** in solvents of different polarity compared with **BP-2TA**. In addition, neat films of **Py-2TA** and **BP-2TA** were fabricated from physical vapor deposition. For both materials, phosphorescent emission is strongly suppressed most likely due to a less rigid molecular environment and triplet-triplet annihilation. As shown in Fig. S34, the emission spectra in aerated and nitrogen atmosphere are almost identical with simultaneous fluorescent and phosphorescent emission and PLQY values of 1.2% for **Py-2TA** and 1.7% for **BP-2TA**.

## Application in PLTs

In addition to the Weber contrast, the quality of PLTs is characterized by their activation behavior (cf. Fig. 1b) over several writing and erasing cycles. For repeated application of the same device, for example, for information storage, the photoluminescence intensity should remain as stable as possible. The activation behavior over the first ten cycles for PLTs made from **Py-2TA**, **PyCN-2TA**, and **BP-2TA** (5 wt% in PS) as a reference is depicted in Fig. 7. **PyCN-2TA** apparently exhibits reduced photostability under UV light illumination, leading to a strong decrease in intensity already after the first activation and even more over ten cycles.

This behavior impedes its application in photonic devices. It is also evident that it is disadvantageous when fluorescence and phosphorescence are close in energy. For **BP-2TA** and **Py-2TA**, the fluorescence signal is further suppressed by the spectral response function of the green channel of the camera (cf. Figs. S36, S37), leading to a very small offset in the activation curves in Fig. 7a when the LEDs are switched on. Correspondingly, for information storage and sensing applications, the fluorescence can be easily filtered to achieve an even better contrast. For **PyCN-2TA**, a significant offset is observed due to its red-shifted fluorescence. According to its increased phosphorescence quantum yield $\varphi_P$, PLTs using **Py-2TA** as emitter achieve a superior maximum intensity after full activation, which is beneficial for applications.

The activation time $t_{act}$ of a PLT is defined by the time required to reach half of its maximum phosphorescence intensity under constant UV-illumination. Accordingly, it is directly proportional to the UV dose applied. In our setup, the activation time for the first activation time of PLTs using **Py-2TA** is $t_{act} = 118 \pm 5$ s and, thus, comparable to those of PLTs using **BP-2TA** with $t_{act} = 111 \pm 5$ s under same UV illumination conditions. Both emitters reveal excellent stability in the maximum intensity over the first ten activation cycles (cf. Fig. 7b). On average, the intensity decreases by 3.5% for **Py-2TA** and by 4.5% for **BP-2TA**. In summary, **Py-2TA** can be considered even more suitable for purely organic photonics solutions based on RTP than **BP-2TA**. This application is not limited to PS as host material. Fully functional PLTs with PMMA as host could also be tested as demonstrated in Fig. S38.

## Conclusions

Digital luminescence as a photonic design motif has the potential to provide cost-efficient yet powerful technological solutions for molecular sensing, bioimaging, anti-counterfeiting, and information storage, with the latter

**Table 2 | Emission properties of BP-2TA, Py-2TA, PyCN-2TA, Pm-2TA, PmCN-2TA in PS at room temperature**

| Compound | $\lambda_{exc}$[a], nm | $\lambda_{max}$ in air[b], nm | $\lambda_{max}$ in $N_2$[c], nm | $K_W$[d] | $\tau_P$[e], ms | PLQY in air/ $N_2$[f], % |
|---|---|---|---|---|---|---|
| **BP-2TA** | 275 | 415 | 520 | 39.1 | 24 | 1/16 |
| | 300 | 455 | 520 | 69.3 | 25 | 1/17 |
| **Py-2TA** | 275 | 440 | 530 | 15.1 | 28 | 2/25 |
| | 300 | 440 | 525 | 16.6 | 29 | 2/28 |
| **PyCN-2TA** | 275 | 480 | 510 | 12.1 | 20 | 2/24 |
| | 300 | 480 | 510 | 13.0 | 20 | 2/22 |
| **Pm-2TA** | 275 | 490 | 500 | 6.5 | 16 | 2/15 |
| | 300 | 490 | 500 | 7.0 | 15 | 2/16 |
| **PmCN-2TA** | 275 | 535 | 535 | 0.7 | 19 | 6/16 |
| | 300 | 535 | 535 | 1.5 | 20 | 7/17 |

[a]Excitation wavelength.
[b]Wavelength of the emission maximum under aerated atmosphere.
[c]Wavelength of the emission maximum under $N_2$ atmosphere.
[d]Weber contrast.
[e]Phosphorescence lifetime extracted from delayed spectroscopy (cf. Fig. 6b, S29b, S30d) via biexponential fits for Py-based compounds and **BP-2TA** or triexponential fits for Pm-based compounds.
[f]PLQY under aerated and $N_2$ atmosphere.

**Fig. 6 | Persistent luminescence of the emitters.** Delayed spectra (**a**) and corresponding phosphorescence decay (**b**) of **Py-2TA**, **PyCN-2TA**, **Pm-2TA**, and **PmCN-2TA** in PS at room temperature under nitrogen atmosphere collected at a delay time of 10 ms, showing only the phosphorescence ($\lambda_{exc} = 275$ nm). Biexponential fit functions (**Py-2TA**, **PyCN-2TA**) and triexponential fit functions (**Pm-2TA**, **PmCN-2TA**) used to extract the phosphorescence lifetimes are presented as red dashed lines in (**b**).

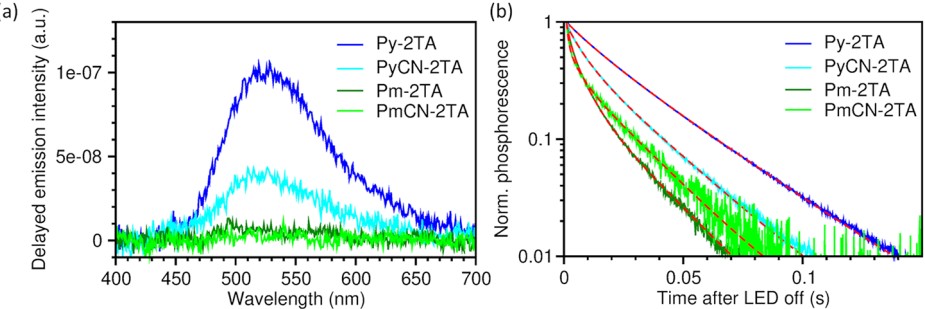

**Fig. 7 | Performance of PLTs using BP-2TA, Py-2TA, and PyCN-2TA as RTP emitters.**
**a** Photoluminescence intensity over illumination time for the first activation cycle. The activation time, where half the maximum phosphorescence intensity is reached, is marked with dashed lines.
**b** Maximum photoluminescence intensity over ten cycles of writing and erasing. In all cases, the intensity values are extracted from the green channel of the camera (cf. spectral response in Fig. S36) and averaged over three PLTs and 420 × 420 pixels à 42.3 × 42.3 μm². The intensity is normalized to the overall highest observed intensity of all materials, which is the maximum photoluminescence intensity of the first cycle of **Py-2TA**. Error bars are omitted for better visibility as no significant variation in the behavior is observed over pixels and PLTs.

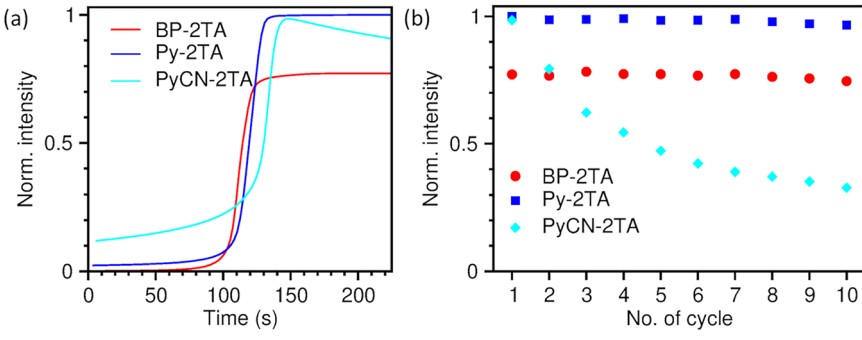

already demonstrated through programmable luminescent tags (PLTs). In this work, we presented a systematic analysis of the impact of the acceptor moiety on the properties of donor-acceptor-donor (D-A-D) RTP emitters, extending the set of available efficient materials and improving the efficiency of PLTs. Although it appears that the electron-accepting character did not systematically modify the photophysical properties, **Py-2TA** emerged as an RTP emitter that outperforms the currently most suitable material **BP-2TA**. **Py-2TA** exhibited significantly improved PLQY in PS films in nitrogen atmosphere, combined with weak yet increased fluorescence and sufficient stability when reused in PLTs for more than ten cycles of writing and erasing. Its strong RTP results from similar properties previously identified for **Pm-2TA**[20] including (a) heavy atom effects and (b) pronounced vibrational spin-orbit coupling. While n-π* transition characteristics are also present for **Py-2TA**, its electronic structure and relevant electronic transitions exhibit reduced charge transfer characteristics and pronounced hybridization, opening an interesting avenue for future RTP emitter design. Despite the variations in chemical and electronic structures of the molecules studied, their RTP emission occurs in a similar wavelength range, peaking between 500 and 535 nm, still limiting the exploitation of the full potential of PLTs and digital luminescence as a design motif.

## Materials and methods
### Materials
3,5-Dibromopyridine (*Sigma-Aldrich*), 3,5-dibromoisonicotinonitrile (*Fluorochem*), 4,6-dibromopyrimidine (*Fluorochem*), 4,6-dibromopyrimidine-5-carbonitrile (*ChemSpace*), thianthrene-1-boronic acid (*Sigma-Aldrich*), tetrakis(triphenylphosphine)palladium(0) (*Thermo Scientific*) and potassium carbonate (*Fisher Scientific*) were used for synthesis of the designed materials. All dry solvents were purchased commercially. Polystyrene ($M_W$ = 35,000 g/mol) obtained from Sigma-Aldrich, Poly(methyl methacrylate) ($M_W$ = 550,000 g/mol) from Alfa Aesar, and Exceval from Kuraray Europe GmbH were used for film fabrication.

### NMR analysis
NMR spectra were measured on a Bruker *AVANCE III HDX, 500 MHz Ascend* (¹H (500.13 MHz), ¹³C (125.75 MHz)). All ¹³C NMR spectra were recorded exclusively with composite pulse decoupling. Chemical shifts were referenced to $\delta_{TMS}$ = 0.00 ppm (¹H, ¹³C). Chemical shifts ($\delta$) are reported in ppm and coupling constants ($J$) are reported in Hz.

### Mass spectrometry
Molecular masses were measured on a TOF/Q-TOF mass spectrometer G6538A by using high-resolution mass spectrometry (HRMS).

### High-performance liquid chromatography (HPLC)
For the estimation of purity of the materials, an Agilent 1260 Infinity II HPLC system with DAD-Detector (254 nm) was used. The sample

concentration was 0.3 mg/mL. Mobile phase A: 30% methanol and mobile phase B: acetonitrile were used with an injection volume of 5 μL.

### X-ray measurements
Suitable single crystals were coated with Paratone-N oil or Fomblin Y25 PFPE oil, mounted using a glass fiber, and frozen in the cold nitrogen stream. X-ray diffraction data were collected at 100 K on a Rigaku Oxford Diffraction SuperNova diffractometer using Cu K$_\alpha$ radiation ($\lambda$ = 1.54184 Å) generated by micro-focus sources. Data reduction and absorption correction was performed using CrysAlisPro[25], respectively. Using Olex2[26], the structures were solved with SHELXT[27] by direct methods and refined with SHELXL[27] by least-square minimization against $F^2$ using first isotropic and later anisotropic thermal parameters for all non-hydrogen atoms. Hydrogen atoms were added to the structure models on calculated positions using the riding model. Images of the structures (cf. Fig. 2) were produced with the Olex2 software.

### Thermal analysis
Simultaneous thermal analysis (STA) was conducted with a *STA 8000* apparatus (Perkin Elmer) under helium gas flow (20 mL/min) and a heating rate of 10 K/min. The "decomposition" temperature was obtained from 5% weight loss and is disclosed as $T_d$.

### Materials simulations
Density functional theory (DFT) and time-dependent DFT (TD-DFT) simulations were performed using the Gaussian 16 package[28]. The long-range corrected hybrid functional CAM-B3LYP[23] was used in combination with the 6-311 G**[21,22] basis set, as this level of theory has shown to accurately predict qualitative trends for the electronic structure and the photo-physical properties of organic materials – partially even at sufficient quantitative accuracy[29,30].

Where available, molecular structures from the crystal structures were used as initial structural guess. Additional conformations were tested to identify the molecule's global energy minimum. The accuracy of the geometry optimization was evaluated based on an analysis of the vibrational spectra for the relaxed structures. All results are based on the conformations found with the lowest total energy. $S_1$ state relaxation was performed using TD-DFT, and $T_1$ state relaxation was performed us unrestricted DFT. Relaxed geometries of ground and excited states are provided in Section S5 in the SI.

By using the relaxed molecular geometry of the neutral molecule in vacuum $q_0$, the vertical ionization potential $IP_{vac}$ and electron affinity $EA_{vac}$ were calculated as

$$IP_{vac} = E_+(q_0) - E_0(q_0) \tag{1}$$

$$EA_{vac} = E_0(q_0) - E_-(q_0) \tag{2}$$

with the ground-state energy $E_0(q_0)$ and the ionic molecular energies $E_\pm(q_0)$. For comparison with experimental cyclovoltammetry (CV) findings, we also calculated the ionization potentials $IP_{CV}$ and electron affinities $EA_{CV}$ in solution with dichloromethane (DCM)[29].

The electron-accepting character of the acceptor moieties (cf. Fig. 1d) was evaluated based on the electrophilicity index $\omega$, calculated from DFT simulations as follows[31]

$$\omega = \frac{\chi^2}{2\eta}. \tag{3}$$

$\chi$ is the electronegativity and $\eta$ the global hardness, which can be obtained from $IP_{vac}$ and $EA_{vac}$

$$\chi = \frac{1}{2}\left(IP_{vac} + EA_{vac}\right) \tag{4}$$

$$\eta = IP_{vac} - EA_{vac}. \tag{5}$$

Note that $\eta$ is sometimes defined by half of the value used here leading to electrophilicity[32] indices half the size given in Fig. 1d.

Absorption spectra were obtained from TD-DFT calculations, including the first 30 excited singlet states. No polarization models were used, as they often do not sufficiently describe absorption in solution or even film. Instead, we represent the excited states by Gaussian functions (broadening of 30 meV) and apply a systematic empirical redshift shift of 500 meV to approximately match the experimental absorption features in PS films. Please mind that the redshift resulting from interaction with the polymer matrix is material-dependent. Our correction scheme was applied to ease comparison between experiment and theory.

### Thin film fabrication for photophysical measurements

1 mL of toluene was added to 15.79 mg of the emitter and 300 mg polystyrene ($M_W = 35{,}000$ g/mol). The mixture was stirred and gently heated until complete dissolution of polymer host and emitter. A volume of 150 μL of solution was used to produce uniform films via spin coating (novocontrol SCE-150) at a speed of 16 rps for 60 s. The film was annealed on a hot plate at 123 °C for 2:15 min.

Neat films of **Py-2TA** and **BP-2TA** were fabricated via thermal evaporation with a Leybold UNIVEX 300 system on quartz glass substrates. The raw materials were first sublimated once. **Py-2TA** was evaporated at 175 °C. **BP-2TA** was slowly and carefully heated to 230 °C, as its material-specific evaporation and decomposition temperatures are close to each other. The deposition was monitored in-situ with a QCM (quartz crystal microbalance) rate control unit CreaPhys RCU001. The rate was kept at 0.35 and 0.45 A/s, respectively, achieving film thicknesses of 72 nm for **Py-2TA** and 78 nm for **BP-2TA**.

### PLTs (programmable luminescent tags) fabrication

1 mL of toluene was added to 15.79 mg of the emitter and 300 mg polystyrene ($M_W = 35{,}000$ g/mol) or 300 mg PMMA ($M_W = 550{,}000$ g/mol). The mixture was stirred and gently heated until complete dissolution. For the oxygen blocking layer, 50 mg Exceval was dissolved in 1 mL of water:IPA 9:1 mixture at 120 °C. For spin coating, a speed of 16 rps for 60 s and volumes of 200 μL of polymer:host and Exceval solutions were used to produce uniform films on a "$1 \times 1$" quartz substrate. Before applying the oxygen barrier layer, the polymer:host films were annealed on a hot plate at 123 °C for 2:15 min.

### UV-vis absorption spectroscopy

The absorbance of individual and blended films spin-coated onto quartz substrates was investigated using a Shimadzu SolidSpec-3700 UV-vis-NIR absorption spectrometer. The absorbance $A$ is calculated as $A = 1 - T - R$, with the transmission $T$ and the reflection $R$ obtained in an integrating sphere.

### Emission measurements

Direct and delayed emission measurements were performed using a CAS 140CTS from Instrument Systems and triggered 275 nm (Thorlabs, M275L4) and 300 nm (Thorlabs, M300L4) LEDs, respectively. For automated data acquisition, the control software SweepMe! was used[33]. All measurements were performed in darkness under nitrogen or aerated conditions. Emission spectra of $10^{-5}$ M solutions of the compounds were measured with instruments mentioned above. For the low temperature measurements at 77 K, the sample was placed in a glass tube filled with liquid nitrogen.

### Phosphorescence lifetime measurements

The phosphorescence lifetime was determined using a silicon photodetector (PDA100A, Thorlabs). The decays were recorded and fitted using a suitable multiexponential fit. The procedure and details can be found in refs. 34, 35.

### Photoluminescence quantum yield

The PLQY values were determined using the method proposed by de Mello et al.[36], improved by F. Fries and S. Reineke[37]. As excitation source, a 300 W xenon lamp combined with a monochromator (LOT Quantum Design MSH300) was used. The samples were placed in a calibrated integration sphere (Labsphere RTC-060-SF) and the spectra acquired with an array spectrometer (CAS 140CT, Instrument Systems).

### Activation curves

The activation curves of PLTs were determined using an experimental setup similar to the one presented by us previously for in-plane oxygen diffusion measurements[38]. The activation behavior of PLTs was recorded under UV-illumination using a CMOS camera (acA1920-40uc, Basler) with focusing lenses (HF25XA-5M, Fujifilm), as shown in Fig. S35. The spectral response curves of the camera sensor are presented in Fig. S36. A 450 nm long-pass filter (FELH0450, Thorlabs) was placed in front of the lens to prevent UV overexposure of the recorded images. Two identical 280 nm (M280L6, Thorlabs) LEDs were used as the excitation source, illuminating the PLTs approximately homogeneously in continuous-wave mode. The mean intensity of the PLT was calculated for each recorded frame. The procedure was controlled using the SweepMe! measurement software[33]. For deactivation, the PLTs were placed on a hot plate at 90 °C for 30 min and then left to rest at room temperature for a further 30 min before the next measurement cycle.

### Activation time

The activation time $t_{act}$ of a PLT was directly extracted from the first activation curve. The offset of this curve can be assigned to the fluorescence intensity $I_F$, while the maximum detected intensity is attributed to the overall photoluminescence $I_L$. The phosphorescent intensity is calculated as $I_P = I_L - I_F$, and the intensity for reading out the activation time is given as $I(t_{act}) = \frac{I_P}{2} + I_F$.

### Data availability

The original contributions presented in the study are included in the article and its Supplementary Information, further inquiries can be directed to the corresponding author. The data that support the findings presented in the figures of the main manuscript are openly available in OpARA at https://doi.org/10.25532/OPARA-646. Crystallographic data for the structures reported in this paper have been deposited at the Cambridge Crystallographic Data Center, under the deposition numbers CCDC 2388219-2388222. Copies of these data can be obtained free of charge via www.ccdc.cam.ac.uk/data_request/cif and are enclosed as Supplementary Data 1-4.

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

## Acknowledgements

Funded by the European Union (ERC, SLOWTONICS, 101089234). Views and opinions expressed are however, those of the author(s) only and do not necessarily reflect those of the European Union or the European Research Council Executive Agency. Neither the European Union nor the granting authority can be held responsible for them. In addition, the research was funded by the European Union's Horizon 2020 Research and Innovation Program under the Marie Skłodowska-Curie grant agreement No. 823720. K.S.S. thanks the Center for Information Services and High Performance Computing (ZIH) at TUD Dresden University of Technology for the use of computational facilities. U.T. thanks the Maria Reiche Welcome Grant of the Graduate Academy (TU Dresden) and the Special Research Fellowship of the Alexander von Humboldt Foundation for financial support. J.F. and J.J.W. thank the TUD Dresden University of Technology, the Graduate Academy of TUD Dresden University of Technology and the German Science Foundation (DFG, WE 4621/6-1) for financial support. The authors thank Dr. Karolis Leitonas for feedback on the manuscript and Dr. Maria James for HPLC measurements (beeOLED GmbH).

## Author contributions

U.T. synthesized the compounds and performed the photophysical characterization. S.K. analyzed the activation behavior of the PLTs. J.F. did the NMR, TGA, DSC and crystallographic analysis. H.T. produced and characterized the PLTs. Y.B.T. contributed to the photophysical characterization. K.S.S. performed the quantum chemical simulations and coordinated the project together with S.R., J.V.G., and J.J.W. All authors contributed to the manuscript.

## Funding

## Competing interests

The authors declare no competing interests.
