## [Transparent Peer Review file · Communications Chemistry]

Systematic variation of the acceptor electrophilicity in donor-acceptor-donor emitters exhibiting efficient room temperature phosphorescence suited for digital luminescence

Corresponding Author: Professor Sebastian Reineke

Version 0:

Reviewer comments:

Reviewer #1

(Remarks to the Author)

The authors first synthesized four novel A-D-A molecules as guests and constructed doped materials using polystyrene as the host. The authors strive to use computational and experimental methods to study the materials. However, the reviewer believes that although the manuscript is suitable for publication, some additional testing is necessary.

- 1, The phosphorescence emission wavelength of the guest molecules in a low-temperature environment (77 K), including solution and solid states.
2. Since the guests are D-A-D type molecules, the authors should supplement the solvent effects of the guest molecules.
3. What are the changes in phosphorescence properties of PS doped with guests of different contents? Authors can simply choose one guest as their research subject.
4. The reviewer understands that under atmospheric conditions, the phosphorescence intensity and phosphorescence lifetime of materials will decrease due to the quenching effect of oxygen. But the authors should explain why there is a significant blue shift in the emission wavelength under the atmospheric condition compared to the nitrogen atmosphere.
5. The reviewer suggests that the authors supplement the SOC and excited state energy levels of the guest molecules, which is beneficial for a deeper understanding of the luminescent properties of the materials.

Reviewer #2

(Remarks to the Author)

This study explores the systematic variation of acceptor electrophilicity in donor-acceptor-donor organic emitters to enhance room temperature phosphorescence (RTP) for advanced photonic applications like digital luminescence. By synthesizing and characterizing compounds based on thianthrene donors and diverse acceptor groups, the research reveals that hybridization within the molecular orbitals significantly improves RTP efficiency and stability. The findings highlight promising materials for programmable luminescent tags with enhanced emission properties, offering sustainable solutions for information storage and sensing technologies. Before the paper is accepted, the author should address the following minor concerns:

1. The ^1H NMR and ^{13}C NMR spectra of compound PmCN-2TA showed obvious impurities that need further purifying.
2. All ESI-MS (m/z) have significant errors, so it is considered that the author should re-test and attach the test results.
3. The CCDC number and check cif report of the crystal needs to be provided.
4. In Figure 6a, the compound corresponding to each line needs to be labeled on the graph.

Reviewer #3

(Remarks to the Author)

In this work, the authors designed and prepared a series of purely organic emitters with donor-acceptor-donor (D-A-D) structure. Notably, BP-2TA in powder exhibited a high phosphorescence efficiency of 21% and high thermal stability. These D-A-D structures in PS demonstrated higher photoluminescence quantum yields in nitrogen compared with that of in the air.

Careful experimental studies, coupled with the theoretical calculations, revealed that room temperature phosphorescence result from heavy atom effects and pronounced vibrational spin-orbit coupling. This work is interesting and could be published on Communications Chemistry after minor revisions.

- 1.The basic characteristic data for the purity of D-A-D compounds should be added, such as HRMS, HPLC etc.
- 2.Additional photophysical properties of purely organic emitters with D-A-D structure in solids should be added, such as, photoluminescence and phosphorescence spectra.
- 3.In previous literatures, poly(methyl methacrylate) was used as the host for programmable luminescent tags. Whether these D-A-D compounds in PMMA is suitable for programmable luminescent tags?

Version 1:

Reviewer comments:

Reviewer #1

(Remarks to the Author)

This reviewer agrees to publish the revised manuscript.

Reviewer #2

(Remarks to the Author)

The points raised in the previous round of review have been satisfactorily addressed. Therefore, I recommend the acceptance of the revised manuscript.

Reviewer #3

(Remarks to the Author)

The revisions have thoroughly resolved my initial concerns. The manuscript is now suitable for publication in Communications Chemistry.

Response to the comments of the Reviewers

Systematic variation of the acceptor electrophilicity in donor-acceptor-donor emitters exhibiting efficient room temperature phosphorescence suited for digital luminescence

Reviewer #1 (Remarks to the Author)

The authors first synthesized four novel A-D-A molecules as guests and constructed doped materials using polystyrene as the host. The authors strive to use computational and experimental methods to study the materials. However, the reviewer believes that although the manuscript is suitable for publication, some additional testing is necessary.

Response:

We thank reviewer #1 for this positive evaluation of our manuscript considering it suitable for publication in *Communications Chemistry*. As described in detail below, we performed several additional experiments and added further explanations to support our findings and promote the quality of the manuscript.

1. The phosphorescence emission wavelength of the guest molecules in a low-temperature environment (77 K), including solution and solid states.

Response:

We agree with reviewer #1 that low-temperature emission spectroscopy can provide further information on the photophysical properties of the emitters. We extended the SI by delayed spectra at 77 K for all emitters in PS films (5 wt%) also depicted below. The spectral shape and emission wavelengths resemble the results obtained at ambient conditions under nitrogen atmosphere supporting the findings presented in Fig. 6 in the main manuscript. However, while for **Pm-2TA** and **PmCN-2TA** delayed spectra could be hardly obtained at ambient conditions, we observed well pronounced phosphorescent emission features at 77 K with emission maxima at expected wavelengths. The delayed spectra for the solutions of all emitters could not be obtained due to limitations in our experimental setup. However, as also characterization of films with reduced emitter concentration shows consistent results (see answer 3), we do not expect further insights from such analyses.

Figure S31. Delayed spectra of **BP-2TA**, **Py-2TA**, **PyCN-2TA**, **Pm-2TA**, and **PmCN-2TA** in PS (5 wt%) at 77 K collected at a delay time of 10 ms showing only the phosphorescence ($\lambda_{\text{exc}} = 275$ nm).

2. Since the guests are D-A-D type molecules, the authors should supplement the solvent effects of the guest molecules.

Response:

Indeed, the effect of solvent polarity on the emission properties of our D-A-D molecules (*solvatochromism*) can provide further insights on the nature of the relevant excited states. We choose our best emitter **Py-2TA** and **BP-2TA** as reference and measured their emission spectra in five different solvents (including toluene, tetrahydrofuran (THF), dimethylformamide (DMF), ethyl acetate, and acetonitrile). The results are presented in **Figure S32** in the SI and a short discussion is added to the main manuscript. They support the findings from materials simulations with the $S_1 \rightarrow S_0$ transition of **BP-2TA** having a pronounced CT state character in contrast to **Py-2TA** with promoted hybridization.

Figure S32. Emission spectra ($\lambda_{exc} = 275$ nm) of **Py-2TA** (a) and **BP-2TA** (b) in solvents of increasing polarity with toluene being least polar and acetonitrile being most polar. For **BP-2TA** the solubility in acetonitrile was not sufficient to measure reliable emission spectra. While **Py-2TA** shows only a minor shift by increasing the polarity of the solvent, **BP-2TA** shows a relatively strong red shift indicating its much more pronounced CT-state character of the $S_1 \rightarrow S_0$ transition.

3. What are the changes in phosphorescence properties of PS doped with guests of different contents? Authors can simply choose one guest as their research subject.

Response:

We thank the reviewer for this valuable comment and extended our study by a photophysical analysis of **Py-2TA**, our best emitter, at 1 and 10 wt% in PS. Emission spectra, delayed spectra, and phosphorescence decays as well as key characteristics are added to the SI and shown also below.

Figure S33. Emission spectra ($\lambda_{exc} = 275$ nm) of pure PS (a) and films of **Py-2TA** (1 wt% (a) and 10 wt% (b)) in PS at room temperature under aerated (dashed lines) and nitrogen atmosphere (solid lines). The pronounced emission peak around 310 nm can be assigned to the PS host. Delayed spectra (c) and corresponding phosphorescence decay (d) under nitrogen atmosphere collected at a delay time of 10 ms showing only the phosphorescence. Biexponential fit functions used to extract the phosphorescence lifetimes are presented as cyan dashed lines in (d).

Table S6. Emission properties of **Py-2TA** in PS at different emitter concentration at room temperature. When the emitter concentration is increased, no significant changes in the photophysical properties are observed, indicating sufficient intermixing and negligible emitter interactions even at elevated concentrations of 10 wt%. Due to relatively pronounced emission by PS in the 1 wt% samples, no reasonable Weber contrast K_W can be defined.

Compound	$\lambda_{exc}^{[a]}$, nm	λ_{max} in air ^[b] , nm	λ_{max} in N ₂ ^[c] , nm	$K_W^{[d]}$	$\tau_P^{[e]}$, ms	PLQY in air/N ₂ ^[f] , %
1 wt% of Py-2TA	275	435	520	-	29	5/23
5 wt% of Py-2TA	275	440	530	15.1	28	2/25
10 wt% of Py-2TA	275	435	520	11.7	27	3/22

^[a] Excitation wavelength. ^[b] Wavelength of the emission maximum under aerated atmosphere. ^[c] Wavelength of the emission maximum under N₂ atmosphere. ^[d] Weber contrast. ^[e] Intensity-weighted phosphorescence lifetime extracted from delayed spectroscopy (cf. Fig. S33d and Fig. 6b) via biexponential fits. ^[f] PLQY under aerated and N₂ atmosphere.

4. The reviewer understands that under atmospheric conditions, the phosphorescence intensity and phosphorescence lifetime of materials will decrease due to the quenching effect of oxygen. But the authors should explain why there is a significant blue shift in the emission wavelength under the atmospheric condition compared to the nitrogen atmosphere.

Response:

We thank reviewer #1 for this question because it helped us to clarify the underlying photophysical phenomena in the main manuscript even more. Under ambient conditions, excited triplet states in purely organic emitters are regularly efficiently quenched by molecular oxygen via triplet-triplet interactions, leading to non-radiative relaxation of the emitter. As a result, we cannot detect any phosphorescence – except for neat films as discussed below (answer 2 to reviewer #3). The emission spectrum only contains fluorescent emission because the excited singlet states are not sensitive to the presence of oxygen. Accordingly, the dashed curves in Fig. 5 in the main manuscript only show the fluorescence of the compounds, which is blue-shifted compared to the phosphorescence due to an elevated S_1 state compared to the T_1 state. Under nitrogen atmosphere, the emission spectrum is dominated by phosphorescence as the emitters are designed to exhibit efficient intersystem crossing and room-temperature phosphorescence. The only exception is **PmCN-2TA**, which has stronger

fluorescence and weaker phosphorescence as well as relatively close emission maxima of both radiative processes.

5. The reviewer suggests that the authors supplement the SOC and excited state energy levels of the guest molecules, which is beneficial for a deeper understanding of the luminescent properties of the materials.

Response:

We fully agree with the reviewer that such information is regularly useful and often provided in publications. We also performed such simulations already before submitting the manuscript, but omitted the results on purpose. Such simulations are usually provided only based on the relaxed S_0 geometry. All our emitter materials – including the reference material **BP-2TA** – are highly flexible also exhibiting further local energy minima that are accessible at room temperature. Already slight modifications in the molecular geometry lead to relatively strong effects on the excited states (NTOs and energies) and the related SOC. This is further promoted by the complex electronic structure of the materials with more than ten excited triplet states below the S_1 state that all need to be considered for comparing, e.g., the ability for intersystem crossing. This means that a presentation of the values only based on the relaxed S_0 geometry will not represent the full characteristics of the material and might even lead to misleading conclusions. Even the choice of the relaxed state is questionable, as the molecules perform a strong relaxation towards the individual excited states including anharmonic effects, which prevents e.g. the simulation of absorption and emission spectra including vibronic features. Therefore, we currently extend the usual computational scheme by an explicit consideration of the molecular dynamics in various energetic states aiming to provide a reasonable representation of the material properties. This goes far beyond the scope of this manuscript and will be presented in a future publication.

Reviewer #2 (Remarks to the Author):

This study explores the systematic variation of acceptor electrophilicity in donor-acceptor-donor organic emitters to enhance room temperature phosphorescence (RTP) for advanced photonic applications like digital luminescence. By synthesizing and characterizing compounds based on thianthrene donors and diverse acceptor groups, the research reveals that hybridization within the molecular orbitals significantly improves RTP efficiency and stability. The findings highlight promising materials for programmable luminescent tags with enhanced emission properties, offering sustainable solutions for information storage and sensing technologies. Before the paper is accepted, the author should address the following minor concerns:

Response:

We thank reviewer #2 for this positive evaluation and are happy to dispel any concerns by extending the experimental analysis and further improving the quality of the manuscript.

1. The ^1H NMR and ^{13}C NMR spectra of compound PmCN-2TA showed obvious impurities that need further purifying.

Response:

As can be seen in the response to reviewer #3 below, we added high-performance liquid chromatography (HPLC) measurements to characterize the purity of our materials. Indeed, **PmCN-2TA** shows slightly reduced purity of 74 %. Still, its characterization shows consistent results. For the most relevant materials within the manuscript, excellent purity beyond 90% is achieved.

2. All ESI-MS (m/z) have significant errors, so it is considered that the author should re-test and attach the test results.

Response:

We agree with the reviewer that the former ESI-MS measurements were not ideal although they did not contain significant errors. We added high-resolution mass spectrometry (HRMS) measurements for further verifying the chemical structures of the molecules.

Figure S3. HRMS spectrum of **Py-2TA**.

MS Zoomed Spectrum

Figure S7. HRMS spectrum of **PyCN-2TA**.

Figure S11. HRMS spectrum of **Pm-2TA**.

Figure S15. HRMS spectrum of **PmCN-2TA**.

3. The CCDC number and check cif report of the crystal needs to be provided.

Response:

Indeed, we forgot both in the submission process. The CCDC numbers are added to the experimental section of the original manuscript. The corresponding CIF reports are submitted along the revised manuscript and SI.

4. In Figure 6a, the compound corresponding to each line needs to be labeled on the graph.

Response:

For a cleaner look of the figure, we originally omitted a legend in Fig. 6a as it uses the same color code and, accordingly, the same legend as Fig. 6b. However, based on the comment of reviewer #2, we modified Fig. 6 and also Fig. S29 as desired.

Figure 6. Persistent luminescence of the emitters. Delayed spectra (a) and corresponding phosphorescence decay (b) of **Py-2TA**, **PyCN-2TA**, **Pm-2TA**, and **PmCN-2TA** in PS at room temperature under nitrogen atmosphere collected at a delay time of 10 ms, showing only the phosphorescence ($\lambda_{exc} = 275$ nm). Biexponential fit functions (**Py-2TA**, **PyCN-2TA**) and triexponential fit functions (**Pm-2TA**, **PmCN-2TA**) used to extract the phosphorescence lifetimes are presented as red dashed lines in (b).

Reviewer #3 (Remarks to the Author):

In this work, the authors designed and prepared a series of purely organic emitters with donor-acceptor-donor (D-A-D) structure. Notably, BP-2TA in powder exhibited a high phosphorescence efficiency of 21 % and high thermal stability. These D-A-D structures in PS demonstrated higher photoluminescence quantum yields in nitrogen compared with that of in the air. Careful experimental studies, coupled with the theoretical calculations, revealed that room temperature phosphorescence result from heavy atom effects and pronounced vibrational spin-orbit coupling. This work is interesting and could be published on *Communications Chemistry* after minor revisions.

Response:

We thank reviewer #3 for this positive evaluation of our manuscript considering it suitable for publication in *Communications Chemistry* with only minor optimization.

1. The basic characteristic data for the purity of D-A-D compounds should be added, such as HRMS, HPLC etc.

Response:

We thank the reviewer for this valuable comment. We used high-performance liquid chromatography (HPLC) to characterize the purity of our materials. The chromatograms and their description are added to the SI. Based on the chromatogram, the purity of each material was estimated: **Py-2TA** (94 %), **PyCN-2TA** (98 %), **Pm-2TA** (91 %), and **PmCN-2TA** (74 %). Accordingly, especially for the most relevant materials **Py-2TA** and **PyCN-2TA** excellent purity could be achieved by column chromatography. Additionally, we provide PL spectra of **Py-2TA** received after recrystallization and after sublimation below within this reply. As can be seen, with even higher purity after sublimation, the photophysical properties are fully consistent.

Figure R1. Emission spectra ($\lambda_{exc} = 275$ nm) of films of **Py-2TA** (5 wt%) at different purity levels in PS at room temperature under aerated (dashed lines) and nitrogen atmosphere (solid lines).

Figure S4. HPLC chromatogram and peak description of **Py-2TA**.

Figure S8. HPLC chromatogram and peak description of **PyCN-2TA**.

Figure S12. HPLC chromatogram and peak description of **Pm-2TA**.

Figure S16. HPLC chromatogram and peak description of PmCN-2TA.

2. Additional photophysical properties of purely organic emitters with D-A-D structure in solids should be added, such as, photoluminescence and phosphorescence spectra.

Response:

We agree with reviewer #3 that photoluminescence spectra in solid state can provide further information on the photophysical properties of the emitters. The neat films of the emitters are obtained by vacuum deposition and presented in Figure S34 in the SI. For both materials, phosphorescent emission is strongly suppressed most likely due to a less rigid molecular environment and triplet-triplet annihilation. The emission spectra in aerated and nitrogen atmosphere are almost identical with simultaneous fluorescent and phosphorescent emission.

Figure S34. Emission spectra ($\lambda_{\text{exc}} = 275 \text{ nm}$) of evaporated neat films of **Py-2TA** (a) and **BP-2TA** (b) at room temperature under aerated (dashed lines) and nitrogen atmosphere and comparison with diluted systems in PS (5 wt%). In neat films, RTP emission already occurs under aerated atmosphere due to a reduced phosphorescence lifetime and, thus, a reduced oxygen sensitivity. The RTP intensity is strongly reduced due to triplet-triplet annihilation and a potentially insufficiently rigid molecular environment.

3. In previous literatures, poly(methyl methacrylate) was used as the host for programmable luminescent tags. Whether these D-A-D compounds in PMMA is suitable for programmable luminescent tags?

Response:

We thank the reviewer in their interest in demonstrating PLTs using PMMA as host. Our best emitter **Py-2TA** was selected for analysis. As can be seen in the Figure below (Fig. S38 in the revised version of the SI), fully functional PLTs can also be obtained by using PMMA as host.

Figure S38. Performance of PLTs using **Py-2TA** as RTP emitter (5 wt%) and PMMA as host. (a) Photoluminescence intensity over illumination time for the first activation cycle. (b) Photo of a locally activated PLT.